# Quantifying non-communicable diseases' burden in Egypt using State-Space model

**Somaya El-Saadani**[1]*, **Mohamed Saleh**[2], **Sarah A. Ibrahim**[1]

**1** Department of Biostatistics and Demography,Faculty of Graduate Studies for Statistical Research, Cairo University, Cairo, Egypt, **2** Faculty of Computers and Artificial Intelligence, Cairo University, Cairo, Egypt

* ssaadani@cu.edu.eg

**Data Availability Statement:** The data of the study are available from public repository. WHO data are available in Global Health Observatory data repository at https://apps.who.int/gho/data/node.main.A867?lang=en. The Annual Health Services

## Abstract

The study aimed to model and quantify the health burden induced by four non-communicable diseases (NCDs) in Egypt, the first to be conducted in the context of a less developing county. The study used the State-Space model and adopted two Bayesian methods: Particle Filter and Particle Independent Metropolis-Hastings to model and estimate the NCDs' health burden trajectories. We drew on time-series data of the International Health Metric Evaluation, the Central Agency for Public Mobilization and Statistics (CAPMAS) Annual Bulletin of Health Services Statistics, the World Bank, and WHO data. Both Bayesian methods showed that the burden trajectories are on the rise. Most of the findings agreed with our assumptions and are in line with the literature. Previous year burden strongly predicts the burden of the current year. High prevalence of the risk factors, disease prevalence, and the disease's severity level all increase illness burden. Years of life lost due to death has high loadings in most of the diseases. Contrary to the study assumption, results found a negative relationship between disease burden and health services utilization which can be attributed to the lack of full health insurance coverage and the pattern of health care seeking behavior in Egypt. Our study highlights that Particle Independent Metropolis-Hastings is sufficient in estimating the parameters of the study model, in the case of time-constant parameters. The study recommends using state Space models with Bayesian estimation approaches with time-series data in public health and epidemiology research.

## Introduction

The epidemiological burden of chronic diseases is increasing worldwide, in the developed countries (DCs) as well as in the less developing countries (LDCs), marking that non-communicable diseases (NCDs) are no longer related to affluence. NCDs are responsible for almost 70% of all deaths worldwide; 85% of these deaths occur in less developing countries [1]. Three main demographic factors drive the noticeable increase in NCDs; aging of the population, population growth, and unplanned urbanization, and other factors such as globalization of unhealthy lifestyles [1, 2]. Egypt, one of the less developing countries where GDP per capita is

Statistical Bulletin are available at https://www.capmas.gov.eg/Pages/Publications.aspx?page_id=5104&Year=23361. The World Bank data were retrieved from World Bank Open data at https://data.worldbank.org/. The data of the institute of Health Metrics and Evaluation are available at http://ghdx.healthdata.org/record/ihme-data/gbd-2017-disability-weights, and http://ghdx.healthdata.org/gbd-results-tool. Data, R codes, and MATLAB codes are attached in supporting file S1 to reproduce the results.

**Funding:** The author(s) received no specific funding for this work.

**Competing interests:** The authors have declared that no competing interests exist.

$3,019 ranking the 132nd [3], is facing a rapid increase in its population size, approaching over 100 million in 2020, with a 2.56% increase rate over the period 2006–2017. Although the share of the older adults aged 60 or above represents 6.7% of the total population in 2017, the old population is increasing faster than the entire population ((intercensal growth rate 3.2 vs 2.40) and amounts over 6 million– far exceeding the size of the older people in most European Union countries [4]. In Egypt, in 2016, 84% of the total number of deaths were due to NCDs, with four groups of diseases accounted for about 60% of the total death, they include; cardiovascular diseases, cancer, chronic respiratory diseases, and Diabetes and kidney diseases (accounted for 40%, 13%, 4%, 3% of the total deaths, respectively) [5]. The probability of premature death induced by NCDs between age 30 and 70 was nearly 28%. Additionally, these four groups of diseases accounted for nearly 44% of the total DALYs in 2019, respectively [6]. Furthermore, the NCDs related risk factors signify an undue load on the health of its adult population. One-third of the adult Egyptian population suffers from high blood pressure, close to one-half of its adult male population smoke tobacco. Rates of physical inactivity, raised cholesterol and obesity are 22%, 23%, and 49%, respectively, among women and 28%, 14% and 25%, among men [7]. Non-communicable diseases (NCDs) and their related risk factors constitute a significant burden over the individuals and the health system in Egypt; where per capita health expenditure is $132 [8], and out-of-pocket health spending is about 60% of the total health expenditure [9]. Although NCDs signify a substantial challenge for socio-economic development, efforts to quantify their burden on the Egyptian population's health are lacking.

During the past few decades, quantifying the disease's burden over the population's health has been a topic of great interest to researchers as well as policymakers. A great deal of research has been conducted in the developed world to quantify the disease burden (communicable and non-communicable) on the population's health. However, such efforts are rare in developing countries [10], and Egypt is no exception. The grand achievement in measuring the population's health status has taken on many forms. Under the umbrella of summary measures of population health (SMPH), some studies assembled information from different health and mortality indicators in one index that reflects the health status of a specific population [11]. Most of these studies used exploratory factor analysis to develop health indices [12–14]. The prime advantage of this method, besides its simplicity, is that it gives one single interpretable value for the individual or the population. However, the disadvantage of using aggregated scores is the inability to know how much information of each domain was included in such a measure [15]. The widely applied type of SMPH, which is based on the life table approach, includes that combined information on fatal and non-fatal health indicators into one comprehensive metric of overall population health [11]. Examples include; active life expectancy (ALE), disability-free life expectancy (DFLE), and disability-adjusted life expectancy (DALYs), with DALYs the primary summary measure of health that is of high usage. These health indices provide internationally standardized measures of populations' health and allow the assessment, evaluation, and monitoring of an individual's or a given community's health status and health-related quality of life. However, they overlook the fact that the average number cannot represent the entire population's health conditions. Most importantly, allocating resources based on the average health index will deepen the inequity between underprivileged and wealthy communities [16]. DALYs faced some additional critiques regarding the assumption of equity of the same disease's burden among varying populations. The severity of the diseases should also be contingent on the social background of subpopulations. Furthermore, the burden should not be alike for developed and underdeveloped countries [17]. Also, in calculating DALYs, the health concept is reduced to the main seven health domains suggested by WHO, which has led to a model specification error. The assumption of independence between severity weights and duration of diseases does not hold [18]. Also, DALYs overlooks the fact that

the availability, quality, and accessibility of healthcare services significantly affect the population's health. The third type of SMPH is based on Multiple Indicator Multiple Cause (MIMIC) models in which health is dealt with as an unobserved construct or latent variable to be determined by its causes and indicators and to be estimated in a system of structural equations. Examples; Multiple Indicator Multiple Cause health Status Index (MIMIC- HSI) [19], and Multiple Indicator Multiple Cause Burden of Disease Index (MIMIC—BDI) [18]. MIMIC-HSI gives more availability for health status with multiple domains. MIMIC-HSI was used to measure the disability caused by some diseases [20]. It was also applied to studies concerned with the population's health and the individual's [21, 22]. The main shortcoming of MIMIC—HSI is the exclusion of non-fatal health outcomes from the index's estimation [23]. Both MIMIC—HSI and DALYs presume the parameters' stability and constant severity weights over a given time [24]. In 2004, Kaltjob proposed (MIMIC—BDI) [18], the disease-related variables were added, and the independence between severity weights and duration of disease was not presumed. Kaltjob used his suggested metric to rank ten different diseases in the year 2000, as well as to investigate their burden on the French population [23]. MIMIC model, however, suffers from several shortcomings. It is found incompetent in circumstances with a small number of observations or observations with absent values [25] and is problematic for applying on time-series data [26, 27]. Additionally, its presumptions of independence between the structural and measurement errors, and the stationary or normally distributed observations are not always applicable [26]. Furthermore, MIMIC's estimated coefficients are not consistent with diverse sample sizes [28]. Although the NCDs' encumbrance in Egypt is substantial, no attempt was conducted to measure their burden. Therefore, the study aims at filling this gap by measuring the NCDs burden trajectories in Egypt. The study's main objective is to develop non-communicable diseases' burden-related health status index (NCDs-BDI) in Egypt. In such an endeavor, health is dealt with as an unobserved (latent) construct characterized by its observable determinants and observable indicators [22]. This effort is the first to be conducted in the context of one of the less developing countries (LDCs), and among the few performed worldwide. Our suggested health metric used the State-Space Model (SSM) to represent the latent health variable's relationship with its causes and indicators. SSM avoids several drawbacks of the Multiple Indicator Multiple Cause (MIMIC-BDI) model [23]. In contrast to the MIMIC, SSM has several advantages. It allows the current state of the latent construct to depend on its previous state and, most importantly, does not impose restrictions on the number of the causes and indicators added to the model [29–31]. SSM is used in studies with a small number of observations. Additionally, it is applied to model time-series observations and studies wherein the number of time points is greater than the number of individual cases. It also allows examining the intra-observations variability [29, 32]. We applied the State-Space Model with two Bayesian methods: Particle Filter (PF) or Sequential Monte Carlo (SMC) method and the Particle Independent Metropolis-Hastings (PIMH) method, and we estimated the burden trajectories of four NCD diseases: 1) cardiovascular diseases, 2) neoplasms, 3) diabetes and kidney diseases, and 4) chronic respiratory diseases. Additionally, we estimated the relationships between the burden and its causes and indicators and compared between the two estimation methods. In composing the non-communicable diseases' burden (NCDs-BDI) index, the study drew on [23][P. 13–16] conceptual framework for population health assessment. In building such a health metric, we conducted some adjustments on the determinants and indicators of the health construct to accommodate better the NCDs' impact (Fig 1, see colored boxes).

Fig 1 summarizes the proposed leading causes and indicators of the disease-related population health index. The supposed determinants include biological and behavioral risk factors, the disease's prevalence, and disease-related disability weight. The study assumed that the

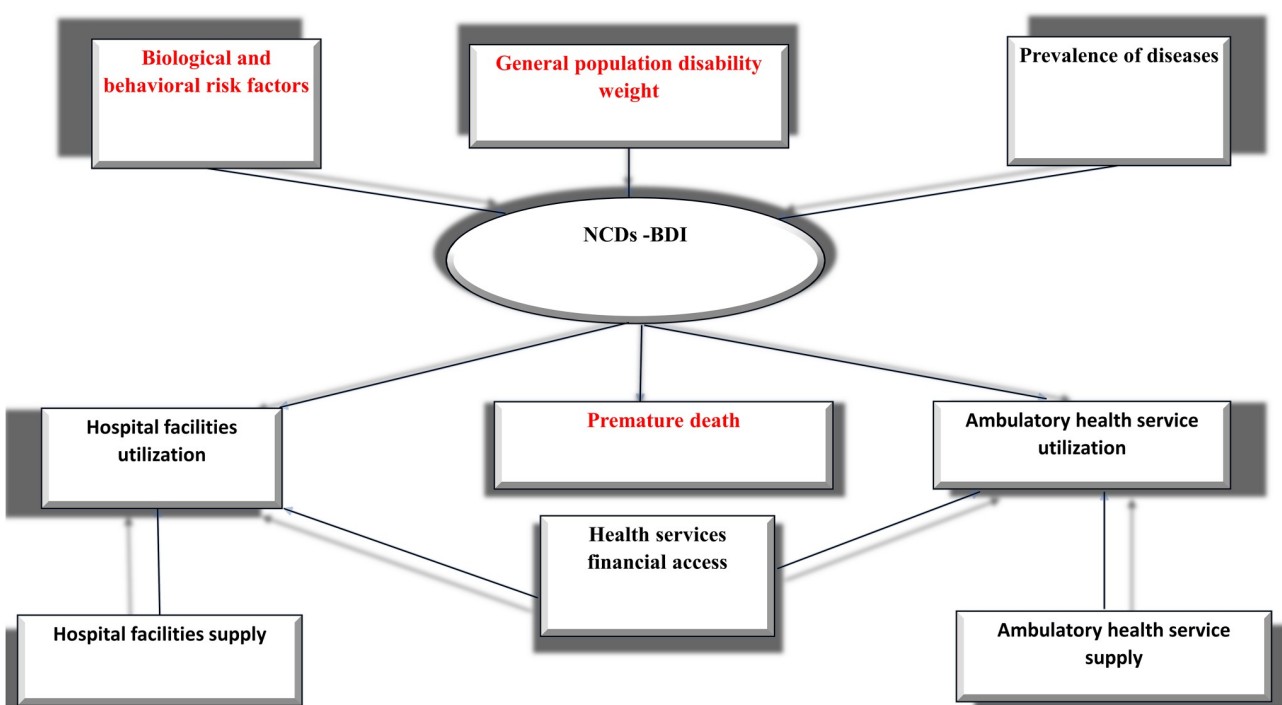

**Fig 1. Conceptual framework for the determinants and indicators of the NCDs burden of diseases.** Source: [18][P. 14]. Note: Modifications by the authors are in colored text in boxes.

biological risk factors (including high blood pressure or high blood glucose) and the behavioral risk factors (such as smoking, unhealthy diet, obesity, alcohol consumption, and physical inactivity) affect health status not only indirectly through increasing the disease incidence [33], but also directly. The risk factors may influence disease burden directly through the behavior of the patients towards their illness. The study assumed that patients with higher risk factors are with low ability to confront their diseases, do not respond quickly to their pains, nor comply with the new therapy, which, in turn, affect the burden of the diseases [34]. The biological and behavioral risk factors were not considered in Kaltjob the framework [18]. The increase in the incidence of a specific pathology causes a rise in the burden of disease-related population's health. Disease-related disability weights are essential determinants of the burden of disease. The study assumed that there is a positive relationship between disability weight and the burden of illness. Indicators of the disease-related population's health metric are presumed to include; premature mortality and health services utilization (hospital facilities utilization and ambulatory health service utilization). Considering premature deaths as an indicator or a cause of disease burden is controversial. In this study, contrary to kaltjob study [18] and in accordance to other studies [35, 36], our framework adopted that premature death is an indicator (i.e., it is a consequence of the population's health status, not a cause). Accordingly, we assumed a positive relationship between the burden of disease and the mortality rates. This assumption is also one of DALYs' main features; the higher the age-specific mortality rates, the worse the health metric [37]. Population's health status, no doubt, is a significant determinant of health service use (both of the hospital facilities and ambulatory health service). Most of the studies agreed that the lower the population health status (i.e., higher disease burden), the higher the utilization of all health services [38–41]. On the other hand, it is not always the case that higher utilization of health services is induced by a higher disease burden. More

heightened awareness of the population can make them quickly respond to their pain. Some studies assumed that increased use of health services might indicate an improvement in population health status because it is associated with therapy and early diagnosis of the diseases or periodical check-ups [42]. Nevertheless, the study assumed that the higher the disease burden, the greater is the utilization of health care services, particularly in a less developing country such as Egypt. Two significant external variables explain health indicators; health care services supply and the population's financial capabilities. Literature supported a positive effect of the availability of health services on its utilization [43]. Moreover, population socio-economic status and its related financial capabilities is a significant determinant of health services utilization. Population socio-economic status is strongly related to the population's awareness and lifestyle, the behavior against the disease symptoms and acute cases, and access to health services [38–40]. In societies such as Egypt, where there is no full health insurance coverage, and individuals' out-of-pocket health expenditure represents about 60% of total health expenditure, financial capabilities positively impact health service utilization, especially if the prices are affordable. Some literature showed a negative relationship between higher prices and utilization of health services [44]. In this study, one of the relationships that were suggested by Kaltjob studies [18, 23], the effect of the disability weights on health services' use, is eliminated as it has no theoretical base. We believe that disability weights indirectly affect the use of health care services through the disease burden. To meet our objective, we organized the study into five sections. Following the introduction, section two displays variables, data sources and their limitations, and details the SSM model and inference methods. Section three delivers the estimates of disease-related population health metrics for four groups of diseases. The discussion and conclusion are provided in section four and five, respectively.

## Materials and methods

### Variables and data sources

The proposed disease-related population health metric NCDs-BDI is estimated for the four groups of NCDs using Egypt macro-level time series data from 1990 to 2017. We used data on disease prevalence rate instead of the incidence rate. It is challenging to find incidence and average duration of disability for all diseases and sequelae [45], notably in LDCs. Estimates of disability weights using population-based surveys have been used as a component of DALYs' measures after 2010 [46]. Therefore, our study as well used the general population disability weights data. Data on the prevalence of each group of diseases (cases per 100,000 population), years of life lost by cause, and the general population disability weights were collected from the Institute for Health Metrics and Evaluation website (IHME) [47]. We used [48] estimates of the prevalence of the five biological and behavioral risk factors: High cholesterol, high blood glucose, high blood pressure, obesity, and smoking. Following [49], we used the number of beds and the number of physicians to measure the supply of health care services: health facilities and ambulatory services. The number of days spent in the hospital and the number of outpatients were used as a proxy of health services (facilities and ambulatory) utilization. Data on these variables were gathered from the Annual Bulletin of Health Services Statistics, Central Agency for Public Mobilization and Statistics, Egypt [50], and were categorized according to physicians' specialities. GDP per capita data are used as a proxy for health services financial access and were gathered from the World Bank's national accounts data [51].

### Data manipulation

**Handling missing data.** Out of the 14 variables, four had missing observations; mainly, the two indicator variables that measure the use of health services (number of days spent in

the hospitals and number of outpatients) and the supply of health services (number of specialists and number of beds). In applying the Particle Filter method of estimation, we used a single imputation method to fill in the missing values for the variables number of specialists and the number of beds. We calculated the averages in this method because the missing data were less than 40% [52]. For the indicators, the number of outpatients and number of days spent in hospitals, we used a technique stemmed from literature [53]. Whenever there was a missing value at time t, they estimated the states based on the available information up to time t-1 [53][Algorithm 2, P. 522]. Inspired by this technique, we used only the available information for each indicator or response variable in time t to calculate the likelihood function; otherwise, the likelihood function of the missing value is considered one and equal weights are assigned to the particles $\left(\frac{1}{N}\right)$ (see Algorithm 1). Accordingly, the likelihood functions were used to calculate the importance weights of the simulated particles according to formula 13, 14, and 15. Assigning a value one to the likelihood function for the missing value will allow us to ignore it in the process of estimating the importance weight as a multiplication of the three weights.

In the instance of applying the Particle Independent Metropolis-Hastings (PIMH), we used a new approach of multiple imputations technique with Amelia package in R [54]. This method has several advantages as it can fit different data mechanisms, keeps the data variability, and gives efficient results in small samples [55]. This new method uses the Expectation-Maximization Bootstrapping approach (EMB). Bootstrapping in Amelia refers to getting several copies from the same dataset and filling them using the expectation-maximization method. Copies of multiple samples ensure the uncertainty in the imputation process. This method uses all the available data, even if it is not used in the analytic model. Multiple imputation gives unbiased estimates and works well with missing at random or missing completely at random data [56]. It is also influential in longitudinal data [57].

Suppose that D is the data matrix, $D \sim (i.i.d.)MVN(\mu, \sigma)$. At first, we assumed initial values for $\mu$ and $\sigma$, then we drew values $(\tilde{D})$ from the assumed multivariate normal distribution (MVN) with these initial values of $\mu$ and $\sigma$ for each copy of the data sets. Afterwards, the expectation-maximization starts. The Expectations is performed using the estimated values of $\mu$ and $\sigma$ (from the previous step) to draw random numbers from the normal distribution to fill in the missing data. Then, we used the complete data to maximize the likelihood function for the two parameters. Iterate until convergence [54]. The likelihood function is $L(\mu, \sigma \mid D) \prod_{i=1}^{N} f_{MVN}(d_i \mid \mu, \sigma), d_i$ is the $i^{th}$ observation. The most conservative assumption in this method is that the data should follow a multivariate normal distribution. If this assumption is relaxed, we can make some transformations to get it as close to normal as possible [54]. In many cases, if we have non-normal or discrete variables, Amelia's normal model works well in imputation [58].

Two steps detected the linear trend of the data. First, we applied the non-parametric Mann-Kendall (MK) test [59, 60] to check the existence of a monotonic upward or downward trend of each series. The null hypothesis of the test assumes that there is no upward or downward trend. It can be applied in case of missing data, but this test doesn't confirm the linearity of the trend. The main advantage of this test, it doesn't require any presumptions of the data distribution. Second, we performed linear interpolation for the missing data, and checked the linearity of the trend to affirm the choice of the linear function in interpolation. We used t-test with Sieve-bootstrap to allow for dependence between observations, assuming that there is no linear trend in the null hypothesis [61, 62]. The following table (Table 1) summarizes the results of the two tests:

**Table 1. Mann-Kendall test and t-test results.**

| Variable | P-value Mann-Kendall | P-value t-test |
|---|---|---|
| Obesity | <.001 | <.001 |
| Tobacco | <.001 | .026 |
| Blood glucose | .009 | <.001 |
| Raised blood pressure | <.001 | <.001 |
| Cholesterol | .002 | .006 |

The results indicate the rejection of the null hypothesis in the two tests, implying the monotonic and linear trend in all of the series. Consequently, it was possible to apply the linear imputation in the five series.

**Algorithm 1** Particle Filter

```
INPUT S₀, Transition equation, Measurement Equation, Observed data
(O_t,1:3)
```

OUTPUT: $\hat{S}_{(t=2:T)} = [\hat{x}, \hat{\theta}_1, \hat{\theta}_2, \hat{\theta}_3, \hat{\theta}_4, \hat{\theta}_5, \hat{\theta}_6, \hat{\theta}_7, \hat{\phi}_1, \hat{\phi}_2, \hat{\phi}_3, \hat{\phi}_4, \hat{\phi}_5, \hat{\sigma}_{1m}^2, \hat{\sigma}_{2m}^2, \hat{\sigma}_{3m}^2, \hat{\sigma}_s^2]$

```
1: Generate iid Particles S₀ ~ P(S₀)
2: for t = 2:T do
3:    η₁:₁₅ ~ N(0, 0.01)
4:    for i = 1:N do
5:       S_it ← Transition equations
6:       y_i1, y_i2, y_i3 ← Measurement equation
7:       e_i,1:3 = O_t,1:3 − y_i,1:3
8:    end for
9:    if y_1t = Nan then
10:       w_i,1t|t−1 = 1
11:    else
12:       Compute w_i,1t|t−1
13:    end if
14:    if y_2t = Nan then
15:       w_i,2t|t−1 = 1
16:    else
17:       Compute w_i,2t|t−1
18:    end if
19:    if y_3t = Nan then
20:       w_i,3t|t−1 = 1
21:    else
22:       Compute w_i,3t|t−1
23:    end if
24:    w_i,1t|t−1 * w_i,2t|t−1 * w_i,3t|t−1 ← w_i,t|t−1
25:    w̃_i,t|t−1 = w_i,t|t−1/sum(w_i,t|t−1) ← normalized − weights
26:    Compute Ŝ_t|t = Σᴺᵢ₌₁ w̃_i,t|t−1 * Sⁱ_t|t−1
27:    Resample
28: end if
```

**Handling high correlation among the five biological and behavioral risk factors.** The five variables that indicate biological and behavioral risk factors, logically, are highly correlated. Therefore, we used the suggested time series factor analysis (TSFA) to collect these variables in one factor that indicates risk factors' prevalence [63]. Time series factor analysis uses the same equation of ordinary exploratory factor analysis but with subscript t. The R package TSFA has been used in this analysis to get the factors that represent the prevalence of risk factors [64]. According to TSFA, we can relax the observation independence and normality; we only need to check if the data are stationary or not and apply differencing if required. Suppose

that at time t, for t equals 1, . . ., T time points, we have k latent variables ($\eta_t$), and M indicators ($y_t$); the model's equation will be as follows [63]:

$$y_t = \alpha + \beta\eta_t + \epsilon_t, \tag{1}$$

where $\alpha$ is M vector of intercept parameters, $\beta$ is M*K matrix of factor loadings, $\epsilon_t$ is M vector of measurement errors. We assumed that the intercept ($\alpha$) is equal to zero in the application of the model [63]. We applied the unit root test Augmented Dickey-Fuller test (ADF) to detect data stationarity [65]. The ADF test depends on the following equation:

$$\Delta y_t = \alpha + \beta t + \gamma y_{t-1} + \delta_1 \Delta y_{t-1} + \delta_2 \Delta y_{t-2} + \cdots + \delta_p + \Delta y_{t-p} + e_t, \tag{2}$$

where $\alpha$ is a constant, $\beta$ the coefficient on a time trend, t is the deterministic trend, and p the lag order of the autoregressive process, and $\Delta y_{t-p}$ is the difference of $p^{th}$ lag order of the series $y_t$. The test detects the null hypothesis of $\gamma=0$. The five variables are non-stationary (each series's mean and variance are not constant and function in time) and should be differenced. Obesity, cholesterol, and blood glucose are integrated of order two. Raised blood pressure and tobacco are integrated of order one (see Table 2).

As we have integrated data of order greater than zero, the mean and variance of indicators will change over time, and the estimation of the constant parameters will be problematic. Consequently, we applied two differences to the five variables to reach stationarity. Then, the equation of the time series factor model will be [63][P. 6]:

$$Dy_t = y_t - y_{t-1} = (\alpha_t - \alpha_{t-1}) + \beta(\eta_t - \eta_{t-1}) + (\epsilon_t - \epsilon_{t-1}), \tag{3}$$

$$Dy_t = \tau_t + \beta D\eta_t + D\epsilon_t. \tag{4}$$

The two extracted factors were assumed to be correlated. The correlation between the two differenced factors was small (0.35). Many methods of rotations can be used in case of interdependent factors such as oblimin, quartimin, geomin, promax, promaj, simplimax, and it is called oblique rotation. Quartimin rotation was used as a rotation method in this analysis [66]. Moreover, we estimated the undifferenced factor scores using Bartlett factor scores to be consistent with the other variables (have the same number of data points), using the following formula [63][P. 12]:

$$\eta_t^\beta = (\beta\prime\omega^{-1}\beta)^{-1}\beta\prime\omega^{-1}y_t. \tag{5}$$

We were able to obtain not time-dependent parameters from the TSFA model using the differenced data series. The resulting Bartlett factor scores depend on the factor loading $\beta$ extracted from the TSFA model 4 and the error covariance $\omega$. [63]. The resulting factor scores were used in the rest of the study.

**Table 2. Augmented Dickey Fuller test results.**

| Variable | Augmented Dickey Fuller test results | |
| --- | --- | --- |
| | P-value before differencing | P-value after differencing |
| Obesity | .98 | .01 |
| Tobacco | .62 | .05 |
| Blood glucose | .98 | .02 |
| Raised blood pressure | .38 | .01 |
| Cholesterol | .71 | .01 |

## Model and statistical analysis

To estimate the latent states' trajectory and the parameters in the State-Space model (SSM), we performed a parallel estimation of the course of the latent states and the parameters using the Bayesian approach. We applied two techniques of the Bayesian approach (we used MATLAB in applying the two methods [67]): Particle Filter (PF) or Sequential Monte Carlo (SMC) method and the Particle Independent Metropolis-Hastings (PIMH) method. The PF assumes that the parameters are dynamic; therefore, we used the online estimation technique in which the estimation is performed sequentially as a new observation is becoming available. In contrast, the PIMH assumes that the parameters are static; hence, we used the offline estimation technique which depends on the entire observations of $y_{1:t}$, y for $t = 1, \ldots, T$ [68].

**Particle filter (sequential Monte Carlo).**    We estimated the latent states' posterior density in the particle filter method based on the observed variables' available information. It is a sequential process of obtaining the latent states' posterior at time t based on the latent posterior at time $t-1$ and the new observed points at time t [69]. Assume that we have the following state equation:

$$x_t = \alpha_1 x_{t-1} + \alpha_2 u_{1t} + \alpha_3 u_{2t} + \alpha_4 u_{3t} + \alpha_5 u_{4t} + \epsilon_{1t}, \epsilon_{1t} \sim N(0, \sigma_s^2) \tag{6}$$

The state equation follows the Markov property; the value of the disease's burden ($x_t$) depends only on the value ($x_{t-1}$). The latent variable also depends on the risk factors ($u_{1t}$), the disease prevalence ($u_{2t}$), the average of mild and moderate disability weights ($u_{3t}$), the average of severe disability weights ($u_{4t}$), and the state noise ($\epsilon_{1t}$). Additionally, we have three measurement equations for the three indicators. The first measurement equation for ($y_{1t}$) refers to the years of life lost due to death (YLL). The indicator variable (YLL) is assumed to be a function in the burden of disease ($x_t$) only, and it takes the form:

$$y_{1t} = \theta_1 x_t + \epsilon_{2t}, \epsilon_{2t} \sim N(0, \sigma_{1m}^2). \tag{7}$$

The second measurement equation is for the ambulatory health services utilization ($y_{2t}$) (proxied by the number of outpatients). It is assumed to be a function in the burden of disease ($x_t$), the ambulatory health services supply (measured by the number of specialists) ($Z_{2t}$), and health services financial access proxied by GDP per capita ($Z_{3t}$) (an estimate of the individual's financial capability); it is written as:

$$y_{2t} = \theta_2 x_t + \theta_3 z_{2t} + \theta_4 z_{3t} + \epsilon_{3t}, \epsilon_{3t} \sim N(0, \sigma_{2m}^2). \tag{8}$$

The third measurement equation is for the indicator ($y_{3t}$), the hospital facilities utilization (proxied by the number of days spent in hospitals). It is assumed to be influenced by the burden of disease, hospital services supply (proxied by the number of beds) ($z_{1t}$), and GDP per capita ($z_{3t}$).

$$y_{3t} = \theta_5 x_t + \theta_6 z_{1t} + \theta_7 z_{3t} + \epsilon_{4t}, \epsilon_{4t} \sim N(0, \sigma_{3m}^2), \tag{9}$$

where $\epsilon_{1t}, \epsilon_{2t}, \epsilon_{3t}$ are the three measurements' noises respectively. For simplicity, we assumed that they follow Gaussian distribution. Regarding the normality assumption of the indicators, we have three response variables. The first response variable, years of life lost (YLL), is a continuous variable due to its calculation methods [70, 71]. The other two response variables (the number of days spent in a hospital and the number of outpatients) have missing values. Therefore, in PIMH, these two response variables needed imputation using EM algorithm to assist random draws of missing values from the normal distribution even if the main distribution of the data is not normal. Consequently, the imputed versions of the data are of continuous type,

and the assumption of normality in PIMH can be acceptable in case of using the imputed data. In particle filter (PF) analysis, we have to choose between two ways of handling count data: either using robust linear models that overcome the shortfalls of the non-normality of the data or make the transformations to approach normality such as log transformation, square root, standardization, Box-Cox transformation [72, 73]. We standardized all the variables to approach normality and have the same assumptions of the normal distribution in the two methods (PF and PIMH) to achieve proper comparison.

The recursive computation of the latent states works as follows [74, P. 139–141]:

According to the Bayes theorem, we can compute the posterior density from the following equation:

$$P(x_t|y_{1:t}) = \frac{P(y_t|x_t)P(x_t|y_{1:t-1})}{p(y_t|y_{1:t-1})}, \tag{10}$$

where the prior density of the latent variable is $P(x_t|y_{1:t-1})$, the likelihood of the data is $P(y_t|x_t)$, and the normalising constant or the marginal likelihood is $p(y_t|y_{1:t-1})$. We discarded the normalising constant for simplicity as it is not always tractable. The previous equation can be rewritten as:

$$P(x_t|y_{1:t}) \propto P(y_t|x_t)P(x_t|y_{1:t-1}). \tag{11}$$

Accordingly, the first step assumes that the initial value of the state, $x_0$ at $t = 1$ follows a density $p(x_0)$. Each iteration t starts with the posterior of $x_{(t-1|t-1)}^{(i)}$ obtained from the previous iteration $t-1$; we can calculate $x_{(t|t-1)}^{(i)}$ from the state (transition) equation to get new samples. We started from $t = 2$, and the estimated states at $t = 1$ are assumed to be the average of the simulated particles from the uniform distribution (Algorithm 1). The importance weight of each particle $w_{t|t-1}^{(i)}$ is calculated according to:

$$w_{t|t-1}^{(i)} = \frac{Target}{Proposal} = \frac{1}{N}\frac{P(x_{t|t-1}^{(i)}|y_{1:t})}{P(x_{t|t-1}^{(i)}|y_{1:t-1})}, \quad \text{for } i = 1, \ldots, N. \tag{12}$$

Where N is the number of particles, $x_{t|t-1}^{(i)}$ are iid samples from $P(x_t|y_{1:t-1})$, and their corresponding weights $w_{t|t-1}^{(i)}$. From Eq 12, as we do not have information about the target distribution we can rewrite the weights as follows:

$$w_{t|t-1}^{(i)} \propto P(y_t|x_{t|t-1}^{(i)}), \quad \text{for } i = 1, \ldots, N. \tag{13}$$

In each iteration, the particles with low weights are discarded, and the new iteration starts with the highly weighted particles. The weights in Eq 13 have three dimensions based on the three measurement Eqs (7, 8 and 9), so we need to estimate $w_{1t}$, $w_{2t}$, $w_{3t}$. The three weight functions are estimated as follows:

$$\hat{w}_{jt}^i = P(y_{jt} \mid x_t, \theta) = \frac{1}{(\sigma_j\sqrt{2\pi})} e^{\frac{-0.5((y_{jt}-\hat{y}_{jt}(\theta'))^2}{\sigma_j^2}}, t = 1, \ldots, T; j = 1, \ldots, 3. \tag{14}$$

According to the previous studies [18, 23], and in agreement with our conceptual framework, the response variables are conditionally independent given the state variable and the parameters (see Fig 1). Assuming that the three indicator variables are independent, we

multiplied the three unnormalized weights following [75]:

$$\hat{w}_t^i = \hat{w}_{1t}^i \times \hat{w}_{2t}^i \times \hat{w}_{3t}^i. \tag{15}$$

The normalized weights (sum to one) are given by:

$$v_t^i = \frac{w_t^{(i)}}{\sum_{i=1}^{N} w_t^{(i)}}. \tag{16}$$

The state estimation was calculated as the average of weighted particles using the following formula:

$$\hat{x}_{t|t} = \frac{1}{N} \sum_{i=1}^{N} x_{t|t-1}^{(i)} w_{t|t-1}^{(i)}. \tag{17}$$

Finally, we added an artificial noise $\psi$ to assume state equations for the parameters, and to allow for the change of parameters through time using a random walk process:

$$\theta_t = \theta_{t-1} + \psi, \psi \sim N(0, 0.01). \tag{18}$$

It should be noted that whatever was the prior distribution of the error term, it would not affect the final results [76]. Using the samples of $x_{1:t}$ we got an unbiased estimator of the likelihood function which was used in the PIMH method according to the following formula [30]:

$$Log\hat{L}(y_{1,1:T}, y_{2,1:T}, y_{3,1:T}; \theta_{1:7}, \alpha_{1:5}, \sigma_1^2, \sigma_2^2, \sigma_3^2)$$

$$= log \prod_{t=1}^{T} \hat{p}(y_{1t}, y_{2t}, y_{3t} \mid x_t, y_{1,1:t-1}, y_{2,1:t-1}, y_{3,1:t-1}, \theta_{1:7}, \alpha_{1:5}, \sigma_1^2, \sigma_2^2, \sigma_3^2) \tag{19}$$

$$= \sum_{t=1}^{T} log \frac{1}{N} \sum_{i=1}^{N} (y_{1t}, y_{2t}, y_{3t} \mid x_{t|t-1}^i, \theta_{1:7}, \alpha_{1:5}, \sigma_1^2, \sigma_2^2, \sigma_3^2).$$

**Particle Independent Metropolis-Hastings.** Particle Independent Metropolis-Hastings method facilitates the inference of the latent state and the parameters of the transition and the measurement equations using Particle Filter (PF) and Markov Chain Monte Carlo (MCMC) simultaneously [77]. It allows for the aligned estimation of the latent states and parameters [30]. As it is an approach for inference by sampling, the parameters are drawn from a proposal density. The main idea is trying to find a Markov Chain for each parameter that converges to a stationary posterior distribution. The principal privilege of this method is that it reaches unbiased estimates [78].

Suppose we have proposal distribution Q, target distribution $\pi$, and proposed or candidate parameter $\theta'$ from proposal distribution $Q(\theta'|\theta^{k-1})$. We should determine if the candidate parameter would be accepted in our chain and becomes $\theta^k$, or stay in the previous position $\theta^{k-1}$. This decision is based on an acceptance probability (The numerator and the denominator contain the posterior distribution $\pi$ so we do not need the normalized constant in this case as it will be cancelled out). $\alpha(\theta', \theta^{k-1})$ [79, P.9-10].

$$\alpha(\theta^{k-1} \rightarrow \theta') = min(1, \frac{\pi(\theta')Q(\theta' \rightarrow \theta^{k-1})}{\pi(\theta^{k-1})Q(\theta^{k-1} \rightarrow \theta')}) \tag{20}$$

$Q(\theta' \rightarrow \theta^{k-1})$ is the transition probability from $\theta'$ to $\theta^{k-1}$, and $Q(\theta^{k-1} \rightarrow \theta')$ is the transition probability from $\theta^{k-1}$ to $\theta'$. If the proposal distribution is a symmetric distribution then, $Q(\theta' \rightarrow \theta^{k-1}) = Q(\theta^{k-1} \rightarrow \theta')$, and the acceptance probability becomes as follows [80]:

$$\alpha(\theta^{k-1} \rightarrow \theta') = \min(1, \frac{\pi(\theta')}{\pi(\theta^{k-1})}). \tag{21}$$

The acceptance probability checks if the new proposed point, under the posterior distribution, is more plausible than the previous one or not. If it is the case, then the acceptance probability would be equal to 1, and the new point will be accepted. Alternatively, we will generate a random number from $U(0, 1)$. If this number is less than the acceptance probability we will accept it with probability $\alpha$; otherwise, the new point will be rejected [79]. The parallel estimation of the latent states and the parameters was a challenge. The combined evaluation will be like two loops: one outer loop for parameters' estimation, and an inner loop for sequential Monte Carlo estimation of the latent states' trajectory and the related likelihood functions based on the estimated parameters in the outer loop (see Algorithm 2) [81]. The unbiased estimator of the likelihood estimated from the particle filter was used in PIMH to calculate the posterior of the parameters (the formula in Eq 19 was used in Algorithm 2, line 7).

In PIMH, the initial points of the parameters were zeros, and that for the measurement and the state's error variances were 0.01. In both techniques, PF and PIMH, we presumed that the coefficients were all positive except for the impact of the diseases' burden on health services' utilization ($\theta_2$ and $\theta_5$), and assumed that the proposal distribution is $U(-1, 1)$. The independent Metropolis-Hastings sampler's positive parameters followed $U(0, 1)$, and the variance of the measurement errors was U(0.1, 1.5). The burden of the disease's initial value follows $U(-3, 3)$.

**Algorithm 2** Particle Independent Metropolis Hastings

```
INPUT: x₀, prior distribution of parameters.
OUTPUT: Estimated Parameters, Estimated states.
```
1: $\theta^0 \leftarrow$ Initialize parameter $\theta_0$
2: $\{\hat{P}_\theta(\hat{y}_{1:T}^{(0)}), X_{1:T}^{(0)}\} \leftarrow$ Particle filter$(\theta^0, X_0)$
3: $\pi^0 \sim \hat{P}_\theta(\hat{y}_{1:T}^{(0)})$
4: **for** i = 2:N **do**
5: $\theta^{i^*} \sim q(\theta^{i^*})$
6: $\{\hat{P}_\theta(\hat{y}_{1:T}), X_{1:T}\} \leftarrow$ Particle filter$(\theta^{i*}, X_0)$
7: $\pi^{i*} \sim \hat{P}_\theta(\hat{y}_{1:T})$
8: Acceptance Probability = min$(1, \frac{\pi^{i*}*prior(\theta^{i*})}{\pi^{i-1}*prior(\theta^{i-1})})$
9: $U \sim U(0, 1)$
10: **if** Acceptance Probability < $U$ **then**
11: $\theta^i \leftarrow \theta^{i*}$
12: $X_{1:T}^i \leftarrow X_{1:T}^{i*}$
13: $\pi^i \leftarrow \pi^{i*}$
14: **else**
15: $\theta^i \leftarrow \theta^{i-1}$
16: $X_{1:T}^i \leftarrow X_{1:T}^{i-1}$
17: $\pi^i \leftarrow \pi^{i-1}$
18: **end if**
19: **end for**
20: Compute the average of each parameter after excluding burn-in iterations: $\hat{\theta}$
21: $X_{1:T} \leftarrow$ Particle filter$(\hat{\theta}, X_0)$

**Table 3. Estimated parameters using particle filter.**

|  | Cardiovascular diseases | Neoplasms | Diabetes and kidney diseases | Chronic respiratory diseases |
|---|---|---|---|---|
| $\alpha_1$ | 0.321 | 0.29 | 0.125 | 0.158 |
| $\alpha_2$ | 0.048 | 0.955 | 0.951 | 0.316 |
| $\alpha_3$ | 0.219 | 0.226 | 0.28 | 0.011 |
| $\alpha_4$ | 0.941 | 0.563 | 0.811 | 0.754 |
| $\alpha_5$ | 0.26 | 0.063 | 0.695 | 0.916 |
| $\theta_1$ | 0.668 | 0.41 | 0.939 | 0.354 |
| $\theta_2$ | 0.217 | 0.18 | -0.152 | -0.272 |
| $\theta_3$ | 0.815 | 0.941 | 0.882 | 0.734 |
| $\theta_4$ | 0.27 | 0.564 | 0.576 | 0.827 |
| $\theta_5$ | -0.817 | 0.251 | -0.406 | -0.381 |
| $\theta_6$ | 0.753 | 0.637 | 0.788 | 0.769 |
| $\theta_7$ | 0.276 | 0.208 | 0.151 | 0.612 |
| $\sigma^2_{1m}$ | 0.043 | 0.112 | 0.189 | 0.093 |
| $\sigma^2_{2m}$ | 0.053 | 0.099 | 0.132 | 0.068 |
| $\sigma^2_{3m}$ | 0.064 | 0.551 | 0.094 | 0.115 |
| $\sigma^2_s$ | 0.538 | 1.254 | 0.366 | 0.368 |

## Results

Table 3 displays the estimated parameters of the state and measurement equations according to the PF method. For each parameter, the estimation converged to a single value, ensuring that the parameters are time-invariant. The computation of the coefficients demonstrates some differences for different diseases.

According to the PF findings, all the diseases' burdens are on the rise, (Fig 2). However, chronic respiratory diseases showed a sharp rise at the beginning of the 1990s, and it leveled off during the time interval 1995 to 2005, then it steeply increased after that. The other three diseases showed a gradual increase with neoplasm revealed a slight rise in its slop after 2005.

The burden of the preceding year weakly predicts the disease burden in the current year for all the diseases ($\alpha_1$). The risk factors ($\alpha_2$) have a strong influence on the burden of neoplasms, chronic respiratory, and diabetes and kidney diseases except for cardiovascular diseases. On the contrary to our hypothesis, the disease prevalence exhibits a low positive impact ($\alpha_3$) on the disease burden of the four groups. The mild/moderate weights apparently affect the burden of the four groups of diseases ($\alpha_4$) and results show a salient positive impact of the severe weights ($\alpha_5$) on the disease burden of chronic respiratory diseases and diabetes and kidney diseases. Considering the health metric's indicators, YLL shows high loadings ($\theta_1$) in cardiovascular, and diabetes and kidney diseases; moderate in neoplasms, and low loading in chronic respiratory diseases. The number of outpatients ($\theta_2$) has low positive loadings in neoplasms and cardiovascular diseases and low negative ones in the remaining two diseases. The number of days spent in hospitals ($\theta_5$) indicates negative loadings in all the diseases, but neoplasms. The number of specialists' effect on the number of outpatients ($\theta_3$) reveals a strong positive impact. GDP per capita ($\theta_4$) has a weak influence on the number of outpatients in cardiovascular diseases, and it shows a strong positive effect in the rest of the diseases. On the other hand, the number beds' influence on the number of days spent in the hospitals ($\theta_6$) was positive strong for the four diseases. Finally, GDP per capita ($\theta_7$) strongly affects the number of days spent in a hospital only for chronic respiratory diseases.

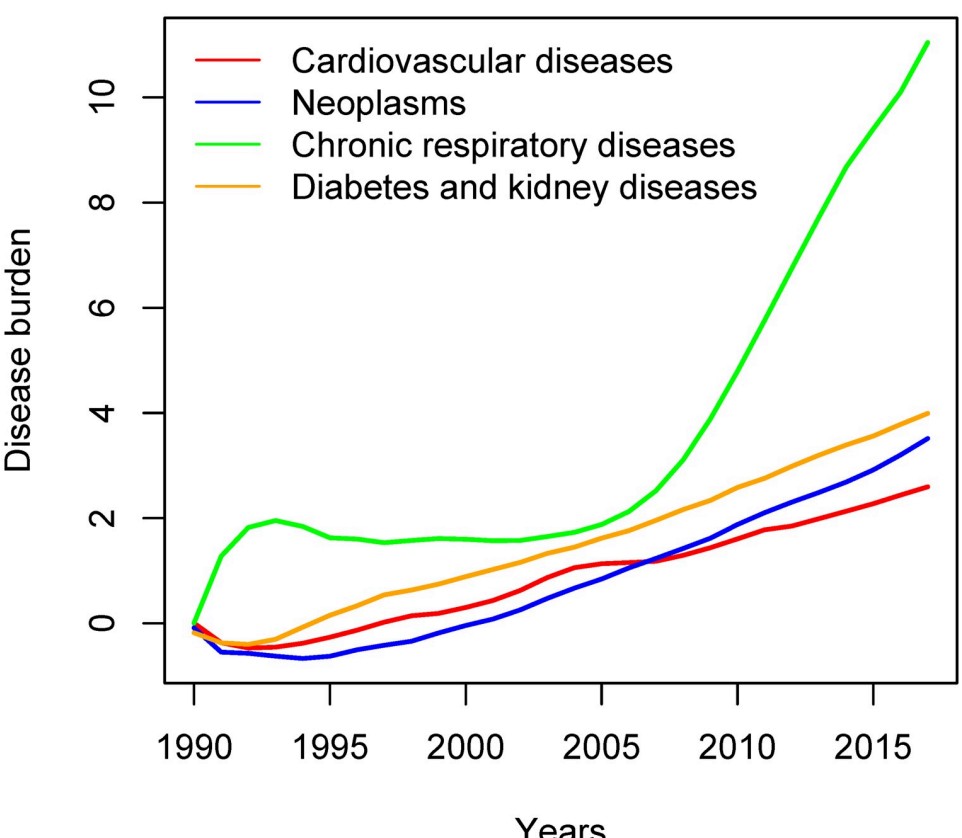

**Fig 2. Particle filter estimated trajectories for the four NCDs group of diseases.**

The estimated error variances of the three measurement equations for the different diseases range between 0.043 (cardiovascular) and 0.189 (diabetes and kidney disease), except for the third measurement error variance in neoplasms (0.551). state equation's error variances are between 0.366 (Diabetes and kidney diseases) and 1.254 (neoplasms).

Regarding PIMH, the resultant estimated parameters were calculated as an average of five imputed samples for each of the four diseases. We used 5000 iterations that were left after discarding the first 1250 burn-in iterations (Burn-in iterations are the first group of iterations that should be discarded from the chain [78]). The number of particles was 1000 in this method [30].

The judgment on the method was performed through several diagnostics. First, the chains' trace plots are stationary around specific values, showing high quality and samples' stability representing the posterior distribution [82] (S1–S4 Figs). The second examines the auto-correlation as an essential indicator of the convergence [81]. We have minimal correlations between the samples and the previous ones (the correlation vanishes after the fourth lag). Also, we have many independent samples that reflect the target distribution (Table in S1 Table). The final diagnostic compares the means of two different segments in the chain; usually the first 10% samples and the last 50% samples. The null hypothesis assumes that the two samples' means are the same [83]. The null hypothesis was accepted, which means that the samples are from the same distribution (p-values of the test are presented in S2 Table). The computation time of PIMH becomes higher, the number of iterations and particles increases (see S3 Table).

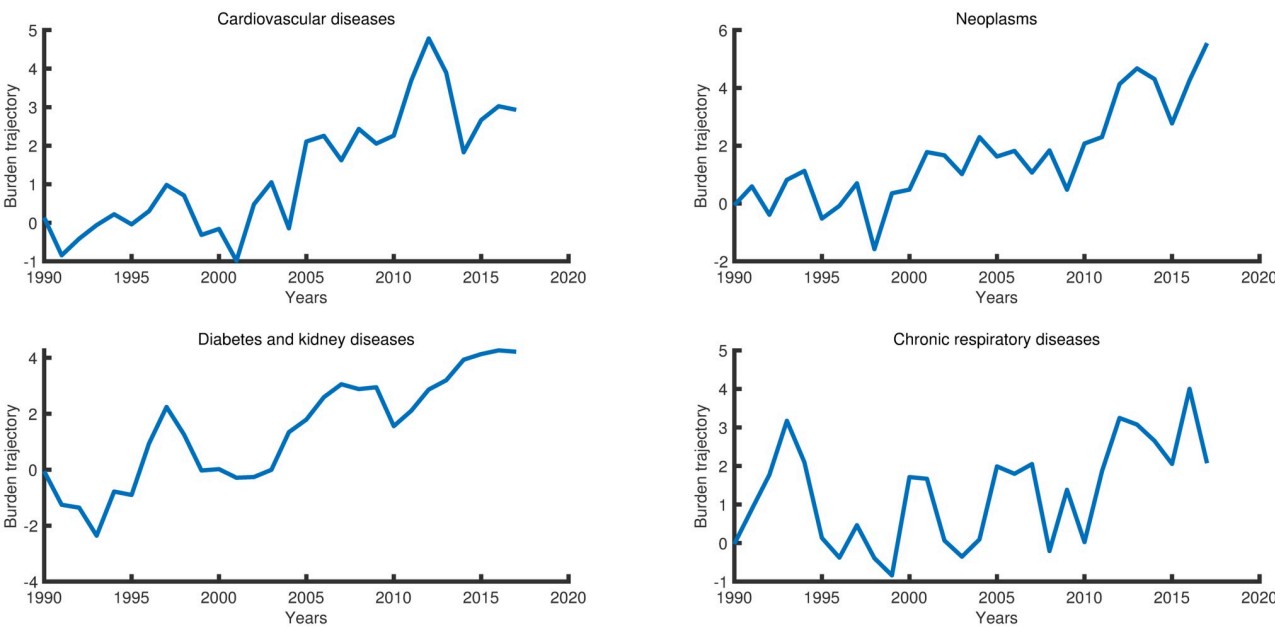

**Fig 3. Particle Independent Metropolis-Hastings estimated trajectories for the four NCDs group of diseases.**

PIMH results revealed an increasing trend of all diseases' burdens (always the first year is affected by the assumed initial values). Cardiovascular, neoplasm, and diabetes and kidney diseases showed a sharp rise, overall, notably after the 2011 revolution. After a sudden decline around 2015, neoplasm kept rising, cardiovascular and chronic respiratory rose then tended to decline, and diabetes and kidney diseases are plateauing after 2015, (Fig 3, and for details, see Fig in S9 Fig).

Regarding NCDs-BDI determinants, results reveal that the rise in the previous year's burden caused an increase in this year's burden with a moderate impact. The higher prevalence of the disease and the prevalence of the risk factors caused a higher disease burden. The same results were reached regarding the moderate impact of the severity weights (Table 4).

Years of life lost due to death (YLL) were significantly affected by diseases' burden (high loading). The average estimated loadings of the number of outpatients ($\theta_2$) are very small and negative in all the diseases (ranges between -0.16 and -0.002). Similarly, the number of days spent in hospitals ($\theta_5$) shows a very weak loading and indicates a negative relationship with the burden for all the diseases but neoplasms (Table 4).

For all diseases, the effects of the specialists' number ($\theta_3$) and GDP per capita ($\theta_4$) on the number of outpatients approached 0.5. On the other hand, the number of beds ($\theta_6$) and the GDP per capita ($\theta_7$) showed a moderately positive effect on the number of days spent in hospitals (around 0.47). Finally, the state and the measurement errors' variances show low variation between diseases 0.7 and 0.8.

## Discussion

The study aimed to develop a burden of disease index in Egypt. In developing such an index, it adopted the conceptualization that it is essential to consider the demand for health care services induced by the disease alongside the disease-related morbidity and mortality. In this endeavor, we adopted the Kaltjob suggested framework [23]. However, we conducted three modifications: (a) We added the biological and behavioral risk factors as one of the major

**Table 4. Estimated parameters using Particle Independent Metropolis-Hastings.**

| | Cardiovascular diseases | | | | Neoplasms | | | | Diabetes and kidney diseases | | | | Chronic respiratory diseases | | | |
|---|---|---|---|---|---|---|---|---|---|---|---|---|---|---|---|---|
| | Mean | SD | 2.50% | 97.50% | Mean | SD | 2.50% | 97.50% | Mean | SD | 2.50% | 97.50% | Mean | SD | 2.50% | 97.50% |
| $\alpha_1$ | 0.401 | 0.272 | 0.025 | 0.95 | 0.382 | 0.264 | 0.021 | 0.934 | 0.423 | 0.285 | 0.014 | 0.962 | 0.422 | 0.283 | 0.019 | 0.944 |
| $\alpha_2$ | 0.469 | 0.293 | 0.024 | 0.985 | 0.469 | 0.291 | 0.024 | 0.963 | 0.483 | 0.291 | 0.015 | 0.98 | 0.474 | 0.284 | 0.026 | 0.979 |
| $\alpha_3$ | 0.474 | 0.275 | 0.03 | 0.964 | 0.466 | 0.284 | 0.023 | 0.958 | 0.47 | 0.283 | 0.03 | 0.956 | 0.463 | 0.278 | 0.036 | 0.964 |
| $\alpha_4$ | 0.457 | 0.292 | 0.008 | 0.976 | 0.447 | 0.283 | 0.018 | 0.956 | 0.469 | 0.283 | 0.021 | 0.952 | 0.46 | 0.287 | 0.019 | 0.968 |
| $\alpha_5$ | 0.452 | 0.28 | 0.014 | 0.963 | 0.438 | 0.286 | 0.03 | 0.954 | 0.461 | 0.284 | 0.019 | 0.954 | 0.454 | 0.278 | 0.019 | 0.965 |
| $\theta_1$ | 0.488 | 0.289 | 0.0192 | 0.982 | 0.517 | 0.292 | 0.021 | 0.982 | 0.482 | 0.277 | 0.031 | 0.964 | 0.49 | 0.277 | 0.035 | 0.979 |
| $\theta_2$ | -0.093 | 0.528 | -0.967 | 0.883 | -0.002 | 0.547 | -0.953 | 0.945 | -0.166 | 0.537 | -0.969 | 0.928 | -0.086 | 0.552 | -0.943 | 0.939 |
| $\theta_3$ | 0.526 | 0.283 | 0.027 | 0.967 | 0.476 | 0.291 | 0.033 | 0.972 | 0.485 | 0.278 | 0.033 | 0.967 | 0.47 | 0.291 | 0.017 | 0.971 |
| $\theta_4$ | 0.47 | 0.287 | 0.022 | 0.978 | 0.52 | 0.293 | 0.029 | 0.982 | 0.476 | 0.289 | 0.019 | 0.968 | 0.494 | 0.286 | 0.019 | 0.967 |
| $\theta_5$ | -0.035 | 0.538 | -0.93 | 0.902 | 0.003 | 0.513 | -0.923 | 0.917 | -0.212 | 0.535 | -0.961 | 0.915 | -0.083 | 0.556 | -0.944 | 0.927 |
| $\theta_6$ | 0.472 | 0.278 | 0.017 | 0.959 | 0.477 | 0.291 | 0.024 | 0.976 | 0.473 | 0.281 | 0.016 | 0.975 | 0.452 | 0.288 | 0.019 | 0.976 |
| $\theta_7$ | 0.463 | 0.283 | 0.014 | 0.974 | 0.505 | 0.279 | 0.028 | 0.98 | 0.458 | 0.289 | 0.026 | 0.98 | 0.469 | 0.288 | 0.025 | 0.973 |
| $\sigma_{1m}^2$ | 0.753 | 0.402 | 0.123 | 1.454 | 0.79 | 0.41 | 0.121 | 1.469 | 0.751 | 0.405 | 0.14 | 1.45 | 0.772 | 0.399 | 0.134 | 1.451 |
| $\sigma_{2m}^2$ | 0.848 | 0.401 | 0.141 | 1.465 | 0.824 | 0.402 | 0.159 | 1.449 | 0.792 | 0.398 | 0.151 | 1.464 | 0.882 | 0.368 | 0.247 | 1.473 |
| $\sigma_{3m}^2$ | 0.854 | 0.383 | 0.172 | 1.469 | 0.788 | 0.399 | 0.158 | 1.461 | 0.878 | 0.387 | 0.197 | 1.471 | 0.882 | 0.377 | 0.193 | 1.46 |
| $\sigma_s^2$ | 0.857 | 0.379 | 0.158 | 1.461 | 0.879 | 0.393 | 0.182 | 1.476 | 0.819 | 0.398 | 0.126 | 1.474 | 0.872 | 0.394 | 0.136 | 1.459 |

causes of the disease burden (were not considered in Kaltjob framework), (b) we considered premature deaths as an indicator (i.e., is a consequence of the population's health status) not a cause as formulated by Kaltjob, and (c) his presumed direct effect of the disability weights on health services' use is eliminated as it has no theoretical base. We believe that disability weights affect the use of health care services indirectly through the disease burden.

This endeavor is the first to be conducted in the context of a less developing county, Egypt, and among the few that had been performed worldwide. The study estimated four burdens of disease indices (NCDs-BDI) for four non-communicable diseases using two Bayesian estimation methods of the State-Space model: The Particle Filter (PF) and the Particle Independent Metropolis-Hastings (PIMH).

The estimated parameters using the Particle Filter method noticeably varied between diseases but static through time, while in the PIMH, the estimated parameters were very close to each other.

Both methods; PF and PIMH came to the conclusion that all the diseases' burdens are on the rise. The slow rise in the burden of neoplasms and chronic respiratory diseases that began at the mid-nineties is most probably influenced by the health sector reform program. The health sector reform program started in 1997 and planned to be accomplished in 2015. It targeted comprehensive coverage of health services and the realization of better health indicators [84, 85]. However, it was obstructed by the aftermath of political instability and economic hardships following the 25th January, 2011 revolution.

Most of the two methods' findings agreed with our assumptions and are in line with the literature. High prevalence of the risk factors, increased disease prevalence, and the increase in the disease's severity level all increase illness burden. The results also showed high loadings for the years of life lost due to death YLL in all the diseases, except for in PF estimation results, YLL has weak loadings for the neoplasms and chronic respiratory. The previous year's burden strongly predicts the current year's burden (the PF results have not confirmed this assumption).

In contradiction to our assumption but in agreement with others [42], the use of health care services had low and negative loadings for all the diseases, except for neoplasms and cardiovascular diseases, suggesting a weak relationship between the burden and utilization. Nevertheless, the results unravel that neoplasms induce hospitalization demand, and cardiovascular induces outpatient clinics' demand. In the more developed countries, health services use can be a matter of high awareness and early check-ups, not a reason for high burden [42]. On the contrary, in Egypt, the negative relationship between disease burden and demand for health care services can be attributed to the lack of full health insurance coverage. Approximately half of the Egyptian population are covered by health insurance. Likewise, half of the retired people that are significantly exposed to chronic diseases have health insurance [86]. Lack of health insurance discourages the use of health care facilities, as it makes seeking health care (visiting doctors for diagnosis, staying at the hospital, buying drugs) costly, and consequently, out-of-pocket spending on health care represents 60% of total health expenditure. Additionally, some sick individuals seek pharmacists' advice and medical prescriptions, a widespread practice in Egypt.

The two methods assure the positive relationship between health services supplies and use, which coincides with our presumption and literature [43]. In accordance with other studies [38, 40, 44, 87, 88], findings show a strong positive relationship between the individual's financial capabilities and health care facilities' use (outpatients), indicating the affluent are more likely to seek health care than the vulnerable and uninsured groups. On the other hand, seeking pharmacists' advice is a major outlet for the poor in case of illness [89]. On the contrary, results reveal a weak relationship between individuals' financial capabilities and hospitalization (inpatients), reflecting that all people, the better-off and the poor, can not escape hospitalization if needed.

It is worth mentioning that although the two methods are different in the assumption of parameters' dynamism; they exhibited a substantial similarity in their findings. It is noteworthy to mention that using PIMH is promising in estimating the parameters and the latent states of the study model, as the parameters converged to a constant value. Diagnostics in PIMH methods assured the convergence to the posterior distribution. On the other hand, in PF, handling missing data in the indicators was much easier than the multiple imputation method applied with PIMH. We performed a sensitivity analysis to assess the performance of the Particle Filter inference method. The Particle Filter method was numerically sensitive to changing the initial values' distribution boundary; it gives estimated parameters of similar directions but with different magnitude. However, it retains the behavior of the latent state. Besides, we used root mean square discrepancy (RMSD) to assess the performance of the varying number of particles (Table in S4 Table). Another difference between the two methods is that PIMH is less numerically sensitive than PF but has greater computation time and (Tables in S3 and S4 Tables). Moreover, we should admit that using PIMH method gave us a higher estimated variance for measurement and state errors.

## Conclusion

The study aimed to estimate an index of the burden of non-communicable diseases on the population's health. This metric will help providing policymakers in Egypt with tools to monitor and forecast future NCDs' progression and model their impact on several dimensions of the societies' demographic and socio-economic development. It also aimed to contribute to the efforts of modeling non-communicable disease trajectories. This attempt is the first to be conducted in Egypt.

The study provided evidence that the burdens of the four NCDs are on the rise. They are positively influenced by their recent past, risk factors, disease prevalence, disability weight, and disease-related deaths.

Our study provided evidence that the State-Space model is a concise representation of the latent variable and its indicators and determinants. The study opens richer insights for the usage of State-Space models with Bayesian estimation approaches in public health and epidemiology instead of ordinary econometric models. Using the previous techniques facilitates the investigation of disease dynamics and simultaneous estimation of latent construct and parameters. The negative and weak relationship between the burden and utilization of health care services found in Egypt's case cannot be generalized. The model should be applied in different countries to assess the assumed relationship between the burden and utilization of health care services in the model. It also highlights the need to enhance Egypt's health registration system as most of the data on demand for health care services are not available and if available, are incomplete.

## Supporting information

**S1 Fig. MCMC sample trace of parameters in cardiovascular diseases.**
(TIF)

**S2 Fig. MCMC sample trace of parameters in neoplasms.**
(TIF)

**S3 Fig. MCMC sample trace of parameters in diabetes and kidney diseases.**
(TIF)

**S4 Fig. MCMC sample trace of parameters in chronic respiratory diseases.**
(TIF)

**S5 Fig. Autocorrelation function in MCMC samples (cardiovascular diseases).**
(TIF)

**S6 Fig. Autocorrelation function in MCMC samples (neoplasms).**
(TIF)

**S7 Fig. Autocorrelation function in MCMC samples (diabetes and kidney diseases).**
(TIF)

**S8 Fig. Autocorrelation function in MCMC samples (chronic respiratory diseases).**
(TIF)

**S9 Fig. Particle Independent Metropolis-Hastings estimated trajectories of the 5000 iterations for the four groups of diseases.**
(TIF)

**S1 Table. Effective sample size for each parameter in Particle Independent Metropolis-Hastings.**
(PDF)

**S2 Table. Geweke diagnostics of the Particle Independent Metropolis-Hastings.**
(PDF)

**S3 Table. Computation time in seconds by inference method.**
(PDF)

**S4 Table. Root mean square discrepancy of each indicator according to number of particles.**
(PDF)

**S1 File. Data and codes' files.**
(ZIP)

## Acknowledgments

The authors would like to thank Dr. Mohamed Fekry Elsayad for his valuable assistance, which helped determine, using the CAPMAS data, the relationship between the disease groups and the physicians' specialties or medical departments.

## Author Contributions

**Conceptualization:** Somaya El-Saadani, Sarah A. Ibrahim.

**Data curation:** Sarah A. Ibrahim.

**Formal analysis:** Mohamed Saleh, Sarah A. Ibrahim.

**Methodology:** Mohamed Saleh, Sarah A. Ibrahim.

**Resources:** Mohamed Saleh.

**Supervision:** Somaya El-Saadani, Mohamed Saleh.

**Visualization:** Sarah A. Ibrahim.

**Writing – original draft:** Sarah A. Ibrahim.

**Writing – review & editing:** Somaya El-Saadani.

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
