## [Decision Letter · Decision Letter 0]

22 Apr 2021

PONE-D-21-00128

Quantifying non-communicable diseases’ burden in Egypt using State-Space model

PLOS ONE

Dear Dr. El-Saadani,

Thank you for submitting your manuscript to PLOS ONE. After careful consideration, we feel that it has merit but does not fully meet PLOS ONE’s publication criteria as it currently stands. Therefore, we invite you to submit a revised version of the manuscript that addresses the points raised during the review process.

We look forward to receiving your revised manuscript.

Kind regards,

Zheng Xu, Ph.D.

Academic Editor

PLOS ONE

Journal Requirements:

Additional Editor Comments:

Please carefully consider the reviewer's comments and prepare the response letter with itemized response together with the revised manuscript. Thanks.

Reviewers' comments:

Reviewer's Responses to Questions

**Comments to the Author**

1. Is the manuscript technically sound, and do the data support the conclusions?

Reviewer #1: Yes

Reviewer #2: Yes

2. Has the statistical analysis been performed appropriately and rigorously? 

Reviewer #1: Yes

Reviewer #2: Yes

3. Have the authors made all data underlying the findings in their manuscript fully available?

Reviewer #1: Yes

Reviewer #2: Yes

4. Is the manuscript presented in an intelligible fashion and written in standard English?

Reviewer #1: Yes

Reviewer #2: Yes

5. Review Comments to the Author

Reviewer #1: The study “Quantifying non-communicable diseases' burden in Egypt using State-Space model” is interesting. In this paper the authors found a negative relationship between disease burden and health services utilization which can be attributed to the lack of full health insurance coverage and the pattern of health care seeking behavior in Egypt. This study highlighted that Particle Independent Metropolis-Hastings is sufficient in estimating the parameters of the study model, in the case of time-constant parameters. The study recommends using state Space models with Bayesian estimation approaches with time-series data in public health and epidemiology research.

The paper is well set, and the problem highlighted executed properly. However, attention should be given to the following highlighted points before resubmitting.

1. In Abstract what is meant by NCD’s and CAPMAS's? Define all abbreviations on their first appearance and then use them simultaneously.

2. Line 201, “Accordingly, we used the likelihood function that has been estimated from the available values of the indicator and assumed the likelihood for the missing values equal to one.” On the basis of what the missing values assume equal 1.

3. Line 211, “The five biological and behavioral risk factors have missing data as well. These variables have shown a linear trend and slight fluctuations. So, we chose the linear interpolation to fill in the missing data.” To check linearity or variability did any test perform?

4. Line 222, “Consequently, two differences were applied to the five variables to reach consistency” how come it possible that every variable is consistent after taking the second differencing.

5. Page 30, Figure 3 may be replaced it is very hard to read the x-axis and y-axis of all four plots.

After all, the paper needs revision and the current form is not acceptable for publication.

Reviewer #2: The authors apply a state-space model into a Bayesian framework to quantify the health burden induced by four non-communicable diseases.

The paper is well written, and the literature well explored. I have some minor concerns about the description of the Bayesian methods.

Pag. 7. The authors describe the particle filter algorithm. The response variables are assumed normally distributed. I am wondering whether this is a plausible assumption for count data.

Formula (5). Please remove the bracket in the denominator.

Line 265. Please substitute "Where" with "where".

Formula (10). The three indicators are assumed independent. Is this a plausible assumption for this study?

Formula (14). The authors claim that the likelihood function is used in the PIMH method. I didn't understand where this formula enters.

6. PLOS authors have the option to publish the peer review history of their article (what does this mean?). If published, this will include your full peer review and any attached files.

Reviewer #1: No

Reviewer #2: **Yes: **Antonino Abbruzzo

---

## [Author Response · Author response to Decision Letter 0]

9 May 2021

Response to reviewers 

PONE-D-21-00128

Quantifying Non-communicable Diseases' Burden in Egypt Using State-Space Model 

PLOS ONE

Reviewer #1: 

1. In Abstract what is meant by NCD's and CAPMAS's? Define all abbreviations on their first appearance and then use them simultaneously.

Edited

2. Line 201, "Accordingly, we used the likelihood function that has been estimated from the available values of the indicator and assumed the likelihood for the missing values equal to one." On the basis of what the missing values assume equal 1.

The likelihood functions were used to calculate the importance weights of the simulated particles according to formula 8, 9, and 10. We used only the available information for each indicator or response variable in time t to calculate the likelihood function; otherwise, the likelihood function of the missing value is considered one. Assigning a value one to the likelihood function for the missing value will allow us to ignore it in the process of estimating the importance weight as a multiplication of the three weights. This idea of handling the missing values in the indicator variables was inspired by Tulsyan, Huang, Bhushan Gopaluni, & Fraser Forbes (2013). Whenever there was a missing value at time t, they estimated the states based on the available information up to time t-1 (Tulsyan, Huang, Bhushan Gopaluni, & Fraser Forbes 2013, Algorithm 2, P.522).

3. Line 211, "The five biological and behavioral risk factors have missing data as well. These variables have shown a linear trend and slight fluctuations. So, we chose the linear interpolation to fill in the missing data." To check linearity or variability did any test perform?

Two steps detected the linear trend of the data. First, we applied the non-parametric Mann-Kendall (MK) test (Hipel & McLeod, 1994; Mann, 1945) to check the existence of a monotonic upward or downward trend of each series. The null hypothesis of the test assumes that there is no upward or downward trend. It can be applied in case of missing data, but this test doesn't confirm the linearity of the trend. The main advantage of this test, it doesn't require any presumptions of the data distribution. Second, we performed linear interpolation for the missing data, and checked the linearity of the trend to affirm the choice of the linear function in interpolation. We used t-test with Sieve-bootstrap to allow for dependence between observations, assuming that there is no linear trend in the null hypothesis (Bühlmann & Buhlmann, 1997; Noguchi, Gel, & Duguay, 2011). The following table summarizes the results of the two tests:

Mann-Kendall test and t-test results:

Variable p-value Mann-Kendall p-value t-test

Obesity <.001 <.001

Tobacco <.001 .026

Blood Glucose .009 <.001

Raised Blood Pressure <.001 <.001

Cholesterol .002 .006

The results indicate the rejection of the null hypothesis in the two tests, implying the monotonic and linear trend in all of the series. Consequently, it was possible to apply the linear imputation in the five series. 

4. Line 222, "Consequently, two differences were applied to the five variables to reach consistency" how come it possible that every variable is consistent after taking the second differencing.

It was a mistake. It should be "Consequently, two differences were applied to the five variables to reach stationarity."

With regard to the second part of the comment, we estimated the undifferenced factor scores using Bartlett factor scores to be consistent with the other variables in the rest of the study (have the same number of data points), as follows:

Time series factor analysis was developed by (Gilbert & Meijer, 2005). He used the same equation of ordinary exploratory factor analysis but with subscript t. The R package TSFA has been used in this analysis to get the factors that represent the prevalence of risk factors. According to TSFA, we can relax the observation independence and normality; we only need to check if the data are stationary or not and apply differencing if required.

Suppose that at time t, for t equals 1,….,T time points, we have k latent variables ( ), and M indicators ( ); the model’s equation will be as follows (Gilbert & Meijer, 2005): 

 , (1)

where is M vector of intercept parameters, is matrix of factor loadings, is M vector of measurement errors. We assumed that the intercept ( ) is equal to zero in the application of the model (Gilbert & Meijer, 2005).

We applied the unit root test Augmented Dickey-Fuller test (ADF) to detect data stationarity (Dickey & Fuller, 1979). 

The ADF test depends on the following equation: 

 = α + βt + , (2)

where α is a constant, β is the coefficient on a time trend, t is the deterministic trend, and p the lag order of the autoregressive process, and is the difference of pth lag order of the series . The test detects the null hypothesis of The five variables were non-stationary (each series's mean and variance are not constant and function in time) and should be differenced. Obesity, cholesterol, and blood glucose are integrated of order two. Raised blood pressure and tobacco are integrated of order one. 

Augmented Dickey Fuller test results

Variable P -value before differencing P-value after differencing

Obesity .98 .01

Tobacco .62 .05

Blood glucose .98 .02

Raised blood Pressure .38 .01

Cholesterol .71 .01

As we have integrated data of order greater than zero, the mean and variance of indicators will change over time, and the estimation of the constant parameters will be problematic. Consequently, we applied two differences to the five variables to reach stationarity. Then, equation (1) of the time series factor model will be (Gilbert & Meijer, 2005, P.6): 

 (3)

 (4)

 The two extracted factors were assumed to be correlated. The correlation between the two differenced factors was small (0.35). Many methods of rotations can be used in case of interdependent factors such as oblimin, quartimin, geomin, promax, promaj, simplimax, and it is called oblique rotation. Quartimin rotation was used as a rotation method in this analysis (Zygmont & Smith, 2014). 

Moreover, we estimated the undifferenced factor scores using Bartlett factor scores to be consistent with the other variables (have the same number of data points), using the following formula (Gilbert & Meijer, 2005, P.12): 

 (5)

 We were able to obtain not time-dependent parameters from the TSFA model using the differenced data series. The resulting Bartlett factor scores depend on the factor loading extracted from the TSFA model and the error covariance . (Gilbert & Meijer, 2005). 

5. Page 30, Figure 3 may be replaced it is very hard to read the x-axis and y-axis of all four plots.

Edited

Reviewer #2: 

Pag. 7. The authors describe the particle filter algorithm. The response variables are assumed normally distributed. I am wondering whether this is a plausible assumption for count data.

In our study, We have three response variables. The first response variable, years of life lost (YLL), is a continuous variable due to its calculation methods (Larson, 2013; Marshall, 2010). The other two response variables (the number of days spent in a hospital and the number of outpatients) have missing values. Therefore, in PIMH, we carried out imputation of the missing values using EM algorithm to assist random draws of missing values from the normal distribution even if the main distribution of the data is not normal. Consequently, the imputed versions of the data are of continuous type, and the assumption of normality in PIMH can be acceptable in case of using the imputed data. 

In particle filter (PF) analysis, we have to choose between two ways of handling count data: either using robust linear models that overcome the shortfalls of the non-normality of the data or make transformations to approach normality such as log transformation, square root, standardization, Box-Cox transformation (Beaujean & Grant, 2016; Zwiener, Frisch, & Binder, 2014). We standardized all the variables to approach normality and have the same assumptions of the normal distribution in the two methods (PF and PIMH) to achieve proper comparison. The following paragraphs detail the imputation process and will be added to the paper.

“ In the instance of applying the Particle Independent Metropolis-Hastings (PIMH), we used a new approach of multiple imputations technique with Amelia package in R (Honaker & King, 2010). This new method uses the Expectation-Maximization Bootstrapping approach (EMB). Bootstrapping in Amelia refers to getting several copies from the same dataset and filling them using the expectation-maximization method. Copies of multiple samples ensure the uncertainty in the imputation process. This method uses all the available data, even if it is not used in the analytic model. Multiple imputation gives unbiased estimates and works well with missing at random or missing completely at random data (Dragset, 2009). It is also influential in longitudinal data (Zhang, 2016).

Suppose that D is the data matrix, . At first, we assumed initial values for and , then we drew values from the assumed multivariate normal distribution ( ) with these initial values for and for each copy of the data sets. Afterwards, the expectation-maximization starts. The expectation is performed using the estimated values of and (from the previous step) to draw random numbers from the normal distribution to fill in the missing data. Then, we used the complete data to maximize the likelihood function for the two parameters. Iterate until convergence (Honaker et al., 2010).

The likelihood function is is the observation. The most conservative assumption in this method is that the data should follow a multivariate normal distribution. If this assumption is relaxed, we can make some transformations to get it as close to normal as possible (Honaker & King, 2010). But in many cases, if we have non-normal or discrete variables, Amelia's normal model works well in imputation (King et al., 2001)."

Formula (5). Please remove the bracket in the denominator.

Edited

Line 265. Please substitute "Where" with "where".

Edited

Formula (10). The three indicators are assumed independent. Is this a plausible assumption for this study?

According to the previous studies (Kaltjob, 2014; Kaltjob, Späth, & Duru, 2004), and in agreement with our conceptual framework, the response variables are conditionally independent given the state variable and the parameters (see Figure 1).

Formula (14). The authors claim that the likelihood function is used in the PIMH method. I didn't understand where this formula enters.

This formula was used in Algorithm 2 (line 7). The unbiased estimator of the likelihood estimated from the particle filter was used in PIMH to calculate the posterior of the parameters.

The following references were added to the list of references in the paper:

Beaujean, A., & Grant, M. (2016). Tutorial on using regression models with count outcomes using R. Practical Assessment, Research, and Evaluation, 21(2), 1–19. https://doi.org/10.7275/pj8c-h254

Bühlmann, P., & Buhlmann, P. (1997). Sieve bootstrap for time series. Bernoulli, 3(2), 123–148. https://doi.org/10.2307/3318584

Dickey, D. A., & Fuller, W. A. (1979). Distribution of the Estimators for Autoregressive Time Series With a Unit Root. Journal of the American Statistical Association, 74(366), 427–431. https://doi.org/10.2307/2286348

Dragset, I. (2009). Analysis of longitudinal data with missing values. Methods and applications in medical statistics (Norwegian University of Science and Technology). Retrieved from https://ntnuopen.ntnu.no/ntnu-xmlui/bitstream/handle/11250/258535/348872_FULLTEXT01.pdf?sequence=2

Hipel, K., & McLeod, A. (1994). Time Series modelling of water resources and environmental systems (1st ed.; A. McLeod, Ed.). Retrieved from https://www.elsevier.com/books/time-series-modelling-of-water-resources-and-environmental-systems/hipel/978-0-444-89270-6

King, G., Honaker, J., Joseph, A., Scheve, K., Advisor, S., Achen, C., … Mccann, J. (2001). Analyzing incomplete political science data: an alternative algorithm for multiple imputation. American Political Science Review, 95(1). Retrieved from http://www.gov.harvard.edu/graduate/tercer/

Larson, B. A. (2013). Calculating disability-adjusted-life-years lost (DALYs) in discrete-time. Cost Effectiveness and Resource Allocation, 11(1), 1–6. https://doi.org/10.1186/1478-7547-11-18

Mann, H. B. (1945). Nonparametric Tests Against Trend. Econometrica, 13(3), 245-259. https://doi.org/10.2307/1907187

Marshall, R. J. (2010). Standard expected years of life lost as a measure of disease burden: An investigation of its presentation, meaning and interpretation. In V. Preedy & W. R. (Eds.), Handbook of Disease Burdens and Quality of Life Measures (pp. 401–413). https://doi.org/10.1007/978-0-387-78665-0_22

Noguchi, K., Gel, Y. R., & Duguay, C. R. (2011). Bootstrap-based tests for trends in hydrological time series, with application to ice phenology data. Journal of Hydrology, 410(3–4), 150–161. https://doi.org/10.1016/j.jhydrol.2011.09.008

Zhang, Z. (2016). Multiple imputation for time series data with Amelia package. Annals of Translational Medicine, 4(3), 56. https://doi.org/10.3978/j.issn.2305-5839.2015.12.60

Zwiener, I., Frisch, B., & Binder, H. (2014). Transforming RNA-Seq data to improve the performance of prognostic gene signatures. PLoS ONE, 9(1), 85150. https://doi.org/10.1371/journal.pone.0085150

Zygmont, C., & Smith, M. R. (2014). Robust factor analysis in the presence of normality violations, missing data, and outliers: Empirical questions and possible solutions. The Quantitative Methods for Psychology, 10(1), 40–55. https://doi.org/10.20982/tqmp.10.1.p040

---

## [Decision Letter · Decision Letter 1]

13 Jul 2021

Quantifying non-communicable diseases’ burden in Egypt using State-Space model

PONE-D-21-00128R1

Dear Dr. El-Saadani,

We’re pleased to inform you that your manuscript has been judged scientifically suitable for publication and will be formally accepted for publication once it meets all outstanding technical requirements.

Kind regards,

Zheng Xu, Ph.D.

Academic Editor

PLOS ONE

Additional Editor Comments (optional):

All comments have been well addressed.

Reviewers' comments:

Reviewer's Responses to Questions

**Comments to the Author**

1. If the authors have adequately addressed your comments raised in a previous round of review and you feel that this manuscript is now acceptable for publication, you may indicate that here to bypass the “Comments to the Author” section, enter your conflict of interest statement in the “Confidential to Editor” section, and submit your "Accept" recommendation.

Reviewer #1: All comments have been addressed

Reviewer #3: All comments have been addressed

2. Is the manuscript technically sound, and do the data support the conclusions?

Reviewer #1: Yes

Reviewer #3: Yes

3. Has the statistical analysis been performed appropriately and rigorously? 

Reviewer #1: Yes

Reviewer #3: Yes

4. Have the authors made all data underlying the findings in their manuscript fully available?

Reviewer #1: Yes

Reviewer #3: Yes

5. Is the manuscript presented in an intelligible fashion and written in standard English?

Reviewer #1: Yes

Reviewer #3: Yes

6. Review Comments to the Author

Reviewer #1: (No Response)

Reviewer #3: The authors have adequately addressed all the comments that raised in a previous round of review

7. PLOS authors have the option to publish the peer review history of their article (what does this mean?). If published, this will include your full peer review and any attached files.

Reviewer #1: No

Reviewer #3: No

---

## [Editor Report · Acceptance letter]

29 Jul 2021

PONE-D-21-00128R1 

Quantifying non-communicable diseases’ burden in Egypt using State-Space model 

Dear Dr. El-Saadani:

I'm pleased to inform you that your manuscript has been deemed suitable for publication in PLOS ONE. Congratulations! Your manuscript is now with our production department. 

Kind regards, 

on behalf of

Professor Zheng Xu 

Academic Editor

PLOS ONE